# Tight First- and Second-Order Regret Bounds
# for Adversarial Linear Bandits

**Shinji Ito**[*]
NEC Corporation
i-shinji@nec.com

**Shuichi Hirahara**
National Institute of Informatics
s_hirahara@nii.ac.jp

**Tasuku Soma**
The University of Tokyo
tasuku_soma@mist.i.u-tokyo.ac.jp

**Yuichi Yoshida**
National Institute of Informatics
yyoshida@nii.ac.jp

## Abstract

We propose novel algorithms with first- and second-order regret bounds for adversarial linear bandits. These regret bounds imply that our algorithms perform well when there is an action achieving a small cumulative loss or the loss has a small variance. In addition, we need only assumptions weaker than those of existing algorithms; our algorithms work on discrete action sets as well as continuous ones without a priori knowledge about losses, and they run efficiently if a linear optimization oracle for the action set is available. These results are obtained by combining optimistic online optimization, continuous multiplicative weight update methods, and a novel technique that we refer to as distribution truncation. We also show that the regret bounds of our algorithms are tight up to polylogarithmic factors.

## 1 Introduction

The *adversarial linear bandit problem* models sequential decision making with limited information, and it has been used in a wide range of applications, including combinatorial bandits [18; 22] and the adaptive routing problem [12]. In this problem, a player is given a set $\mathcal{A} \subseteq \mathbb{R}^d$ of actions represented by $d$-dimensional feature vectors, and the player is to choose an action $a_t \in \mathcal{A}$ in each round $t = 1, \ldots, T$ of the decision process. Just after choosing the action $a_t$, the player receives the *bandit feedback* $\langle \ell_t, a_t \rangle$ as the loss of the action, where $\ell_t \in \mathbb{R}^d$ is the *loss vector* of the $t$-th round chosen by an adversary.[1] We should note here that the loss vector $\ell_t$ is not revealed to the player even after choosing the action. The goal of the player is to minimize cumulative loss $\sum_{t=1}^{T} \langle \ell_t, a_t \rangle$. Player performance of the player is evaluated by means of the *regret* $R_T(a^*)$ defined as

$$R_T(a^*) = \sum_{t=1}^{T} \langle \ell_t, a_t \rangle - \sum_{t=1}^{T} \langle \ell_t, a^* \rangle \tag{1}$$

for $a^* \in \mathcal{A}$. A plethora of algorithms have been proposed with the expected regret $\mathbf{E}[R_T(a^*)] = \tilde{O}(d\sqrt{T})$ for any $a^* \in \mathcal{A}$ [18; 32; 40] under the assumption that $|\langle \ell_t, a_t \rangle| = O(1)$, where $\tilde{O}(\cdot)$ hides a logarithmic factor in $d$ and $T$. These algorithms are *worst-case optimal* up to logarithmic factors:

---

[*]This work was done while Shinji Ito was at the University of Tokyo.

[1]In this paper, we are concerned with *adaptive adversaries*, i.e., an adversary can choose loss $\ell_t$ based on the past player's actions $a_1, \ldots, a_{t-1}$. We note that $\ell_t$ cannot depend on the $t$-th action $a_t$ since otherwise the player would always suffer $\Omega(T)$ regret.

Table 1: Regret bounds for adversarial linear bandits. Here, we denote $\bar{\ell} = \frac{1}{T}\sum_{t=1}^{T}\ell_t$. The results with † require additional assumptions on the feasible set $\mathcal{A}$ and a priori knowledge of deviations.

|  | Upper Bound | Lower Bound |
|---|---|---|
| Worst case | $\tilde{O}(d\sqrt{T})$ [18; 32] | $\Omega(d\sqrt{T})$ [25; 35] |
| First order | $\tilde{O}\left(d\sqrt{\sum_{t=1}^{T}\langle\ell_t, a^*\rangle}\right)$ **[Theorem 3]** | $\Omega\left(d\sqrt{\sum_{t=1}^{T}\langle\ell_t, a^*\rangle}\right)$ |
| Second order | $\tilde{O}\left(d\sqrt{\theta\sum_{t=1}^{T}\|\ell_t - \bar{\ell}\|_2^2}\right)$ † [30] $\tilde{O}\left(d\sqrt{\sum_{t=1}^{T}\|\ell_t - \bar{\ell}\|_*^2}\right)$ **[Theorem 2]** | $\Omega\left(d\sqrt{\sum_{t=1}^{T}\|\ell_t - \bar{\ell}\|_*^2}\right)$ |
| Predictable sequence | $\tilde{O}\left(d\sqrt{\theta\sum_{t=1}^{T}\|\ell_t - m_t\|_2^2}\right)$ † [44] $\tilde{O}\left(d\sqrt{\sum_{t=1}^{T}(\langle\ell_t - m_t, a_t\rangle)^2}\right)$ **[Theorem 1]** | $\Omega\left(d\sqrt{\sum_{t=1}^{T}\|\ell_t - m_t\|_*^2}\right)$ |

Any algorithm suffers the expected regret of $\Omega(d\sqrt{T})$ in the worst case [25; 35]. These worst-case bounds are, however, too pessimistic in practical situations since we rarely encounter truly adversarial environments in real-world problems.

To get around the worst-case lower bound, algorithms with *first-order* and *second-order* regret bounds have been developed for some adversarial bandit problems [6; 15; 30; 48]. First-order regret bounds are those depending on the minimum cumulative loss $L_T^* = \min_{a^*\in\mathcal{A}}\sum_{t=1}^{T}\langle\ell_t, a^*\rangle$, rather than on the number $T$ of rounds. For example, Allenberg et al. [6] proposed an algorithm with a first-order regret bound for the *adversarial multi-armed bandit (MAB) problem*, a special case of the adversarial linear bandit problem in which the action set $\mathcal{A}$ is just a finite set of size $K$.[2] For MAB, their algorithm achieves regret of $\tilde{O}(\sqrt{KL_T^*})$, which improves over the worst-case optimal bound of $O(\sqrt{KT})$ [8; 11], especially when $L_T^*$ is much smaller than $T$, i.e., where there is an action with a small cumulative loss. It is worth noting that their algorithm achieves a nearly optimal *worst-case* bound, as well, since $L_T^* \leq T$ follows from a standard assumption. For more general linear bandits, however, such an algorithm was not known in the literature. In this paper, second-order regret bounds refer to those depending on the *second-order variation* $\sum_{t=1}^{T}\|\ell_t - \bar{\ell}\|^2$ rather than on $T$, where $\|\cdot\|$ is an arbitrary norm and $\bar{\ell}$ stands for the average of the loss vectors $\{\ell_t\}_{t=1}^{T}$.[3] For MAB, Bubeck et al. [15] have proposed an algorithm with a regret bound of $\tilde{O}(\sqrt{\sum_{t=1}^{T}\|\ell_t - \bar{\ell}\|_2^2})$. This algorithm is nearly worst-case optimal since $\|\ell_t - \bar{\ell}\|_2^2 \leq KT$, and it performs better when losses $\{\ell_t\}$ have small variation. For linear bandits, Hazan and Kale [30] proposed an algorithm achieving $\tilde{O}(d^{\frac{3}{2}}\sqrt{\sum_{t=1}^{T}\|\ell_t - \bar{\ell}\|_2^2})$ regret under certain assumptions. Though this bound is better than the worst-case optimal bound of $\tilde{O}(d\sqrt{T})$ when $\sum_{t=1}^{T}\|\ell_t - \bar{\ell}\|_2^2 = O(T/d)$, it is not worst-case optimal in general.

Another (and deeply relevant) line of work that tries to get around the worst-case lower bound is a framework called *predictable sequences* [44; 45; 48]. This framework assumes that the player is given *predicted loss vector* $m_t$, which is produced by an arbitrary process, before choosing actions. One would hope that the regret would get smaller when $m_t$ predicts $\ell_t$ well. In fact, as shown in [44], we can achieve $\tilde{O}(d^{\frac{3}{2}}\sqrt{\sum_{t}^{T}\|\ell_t - m_t\|_2^2})$ regret for linear bandits, which can be smaller than the worst-case optimal bound if $m_t$ are sufficiently close to $\ell_t$ so that $\sum_{t=1}^{T}\|\ell_t - m_t\|_2^2 = O(T/d)$.

**Our contributions**

In this paper, we present newly-devised, efficient algorithms with improved first- and second-order regret bounds for the adversarial linear bandit problem. Our algorithms not only yield better regret bounds but also make only fewer assumptions than have been seen in previous studies. Previous results and our contributions are summarized in Table 1. In the table, $\bar{\ell}$ denotes the average of the loss vectors, i.e., $\bar{\ell} = \frac{1}{T} \sum_{t=1}^{T} \ell_t$. The bounds with † in Table 1 require additional prior knowledge w.r.t. the loss vectors. For example, the results by [30] and [44] in Table 1 are based on the assumption that, respectively, (approximated values of) the quantities $\sum_{t=1}^{T} \|\ell_t - \bar{\ell}\|^2$ and $\sum_{t=1}^{T} \|\ell_t - m_t\|_2^2$ are given before the game starts. In addition, they assume the following conditions: (i) The action set $\mathcal{A}$ is convex, (ii) $\max_{a \in \mathcal{A}} \|a\|_2 = O(1)$, and (iii) $\mathcal{A}$ has a self-concordance barrier with parameter $\theta \geq 1$. Note that the self-concordance parameter $\theta$ can be $\Omega(d)$, e.g., when $\mathcal{A} = \{a \in \mathbb{R}^d \mid \|a\|_\infty \leq 1\}$.

We provide two algorithms with the guarantees described below:

1. The first one achieves $\mathbf{E}[R_T(a^*)] = \tilde{O}\left(\mathbf{E}\left[d\sqrt{\sum_{t=1}^{T} (\langle \ell_t - m_t, a_t \rangle)^2}\right]\right)$ for predictable sequences. If $\|a_t\| = O(1)$ holds for a fixed norm $\|\cdot\|$, our bound implies a regret bound of $\tilde{O}\left(\mathbf{E}\left[d\sqrt{\sum_{t=1}^{T} \|\ell_t - m_t\|_*^2}\right]\right)$, where $\|\cdot\|_*$ is the dual norm of $\|\cdot\|$. This result encompasses the result by [44] in Table 1 since $\max_{a \in \mathcal{A}} \|a\|_2 = O(1)$ is assumed in their work and $\theta \geq 1$ in general. Further, this algorithm does not require the above-mentioned assumptions. In particular, the action set can be discrete.

2. The second achieves the following second-order regret bound: $\mathbf{E}[R_T(a^*)] = \tilde{O}\left(d\sqrt{\sum_{t=1}^{T} \|\ell_t - \bar{\ell}\|_*^2}\right)$ for arbitrary $\bar{\ell}$, as shown in Theorem 2. This result encompasses the regret bound by [30] in Table 1, and again, as before, this algorithm does not require the above-mentioned assumptions. When losses $\langle \ell_t, a_t \rangle$ are non-negative, this algorithm achieves the first-order regret bound of $\mathbf{E}[R_T(a^*)] = \tilde{O}\left(d\sqrt{\sum_{t=1}^{T} \langle \ell_t, a^* \rangle}\right)$ simultaneously, as shown in Theorem 3.

Each regret bound shown in Theorems 1, 2 and 3 is of $\tilde{O}(d\sqrt{T})$, and hence, enjoys worst-case optimality up to logarithmic factors, in contrast to existing algorithms [30; 44]. We should note, however, that our regret bounds are not tight for MAB since the worst-case optimal regret bound is known to be $\Theta(\sqrt{dT})$ for this special case. Similarly, in the special case where the action set is the unit ball, algorithms proposed in [30; 44] achieves regret bounds comparable to ours (up to logarithmic factors), as there is a self-concordant barrier of parameter $\theta = O(1)$.

Our algorithms are based on the multiplicative weight update (MWU) method [7; 34] with an unbiased estimator $\hat{\ell}_t$ of the loss vector $\ell_t$. As with existing algorithms [18; 30; 32] for the adversarial linear bandit problem, we construct an unbiased estimator from a single observation $\langle \ell_t, a_t \rangle$, where $a_t$ follows a distribution $p_t$ maintained by the MWU method. The regret strongly depends on the *stability* of the unbiased estimators $\hat{\ell}_t$; we want the norm and variance of $\hat{\ell}_t$ to be small enough. In order to make an unbiased estimator stable, previous studies [18; 32] have mixed $p_t$ with another probability distribution. This approach, however, requires the mixing rate of $\Omega(d/\sqrt{T})$, which causes $\Omega(T \cdot d/\sqrt{T}) = \Omega(d\sqrt{T})$-regret in general. To overcome this issue, we *truncate* (the support of) the distribution $p_t$ instead. More specifically, we truncate the distribution to ensure that the magnitude of chosen actions would be controlled with respect to an appropriately designed norm. We will show here that this approach ensures the stability of $\hat{\ell}_t$ with almost no degradation of the expected performance, with the help of a concentration property of log-concave distributions [42]. We should note that similar techniques of truncating distributions can be found in the literature of bandit optimization, such as combinatorial semi-bandits [43] and bandit convex optimization [14]. This paper, however, employs a different way of truncation and analyses as the problem settings are different.

Another essential element in our algorithms is the technique called *optimistic online optimization* [44; 45]. Rakhlin and Sridharan [44; 45] introduced the framework of online optimization with predicted

loss $m_t$, and proposed algorithms referred to as *optimistic online mirror descent* and *optimistic follow the regularized leader* that exploit the predicted loss $m_t$ to improve the regret. Our first algorithm employs their techniques to achieve the regret in Theorem 1. The second appropriately chooses $m_t$ on the basis of an online optimization method, and it achieves regret bounds noted in Theorems 2 and 3. A similar technique for computing $m_t$ can be found in the study by Cutkosky [23], though this existing work uses a different loss as it is for the full-information setting.

## 2 Related Work

The linear bandit problem generalizes, as a special case, the well-studied multi-armed bandit (MAB) problem [11], in which the action set $\mathcal{A} = [K] := \{1, 2, \ldots, K\}$ is just a finite set of actions. Other important special cases are *combinatorial bandits* [13; 18], in which the action set $\mathcal{A} \subseteq 2^{[K]}$ is a family of subsets of a finite set, and the loss incurred by choosing an action $a \in \mathcal{A}$ is $\sum_{i \in a} \ell_{ti}$. For example, given a directed graph $G = (V, A)$ and nodes $s, t \in V$, by setting $\mathcal{A}$ to be the family of edge sets representing $st$-paths, one can model the *bandit shortest path* or *adaptive routing* problem [12] as a combinatorial bandit. Further, online recommendation problems have been formulated as linear bandits [41]. To solve linear bandits, many algorithms have been proposed for stochastic settings [1; 10; 21], as well as for adversarial settings [2; 3; 12; 13; 18; 32]. It was shown that we can achieve $\tilde{O}(d\sqrt{T})$ regret [13; 20], which nearly matches the lower bound of $\Omega(d\sqrt{T})$ shown in [25]. The truncation technique in this paper has recently been applied to the delayed-feedback setting as well [37].

A seminal work by Freund and Schapire [26] provided a first-order regret bound for the *expert problem*, a full-information counterpart of MAB. For MAB, Allenberg et al. [6] proposed an algorithm with a first-order bound. This result has been extended in two directions. First one is the *contextual bandit problem* [11; 39], in which the player is given a *context* $x_t \in X$ before choosing the action and the regret is measured by means of a *hypothesis set* $\Pi \subseteq \{\pi : X \to [K]\}$ as $R_T = \sum_{t=1}^{T} \ell_{ta_t} - \min_{\pi^* \in \Pi} \sum_{t=1}^{T} \ell_{t\pi^*(x_t)}$. Offering a first-order regret bound for the contextual bandit was posed as an open problem [4], and Allen-Zhu et al. [5] solved it affirmatively. Another direction is combinatorial semi bandits [9; 28; 47], a variant of combinatorial bandits with more informative feedback, in which the player who chose $a_t \in \mathcal{A} \subseteq 2^{[d]}$ can observe loss $\ell_{ti}$ for each $i \in a_t$. For combinatorial semi bandits, Neu [43] proposed an algorithm with a first-order regret bound. This algorithm, however, does not apply directly to the full-bandit setting in which only $\sum_{i \in a_t} \ell_{ti}$ is observable.

The notion of second-order regret bound was introduced by Cesa-Bianchi et al. [19] for the (full-information) expert problem, in which a regret bound of $O(\sqrt{\log K \cdot Q^*})$ where $Q^* \leq \max_{a \in [K]} \sum_{t=1}^{T} \ell_{ta}^2$ is known beforehand. Hazan and Kale [29] improved this result by replacing $Q^*$ with the *variation* $V^* \leq \max_{a \in [K]} \sum_{t=1}^{T} (\ell_{ta} - \bar{\ell}_{ta})^2$ of the loss sequence.[4] For MAB, Hazan and Kale [30] proposed an algorithm achieving $\tilde{O}(K^2\sqrt{V})$-regret with $V = \sum_{t=1}^{T} \|\ell_t - \bar{\ell}\|_2^2$, and they conjectured that there exists an efficient algorithm with an $\tilde{O}(\sqrt{V})$-regret bound [31]. Bubeck et al. [15] proved this conjecture by providing such a regret upper bound, which almost matches a lower bound of $\Omega(\sqrt{V})$ provided by Gerchinovitz and Lattimore [27]. Wei and Luo [48] provided an MAB algorithm with $\tilde{O}(\sqrt{KS})$ regret, where we denote $S = \sum_{t=1}^{T} (\ell_{ta^*} - \bar{\ell}_{ta^*})^2$ for $a^* \in \arg\min_{a \in [K]} \sum_{t=1}^{T} \ell_{ta}$, which is incomparable to $\tilde{O}(\sqrt{V})$ in general. It is worth noting that MAB algorithms mentioned here require a priori knowledge of parameters $V$ and $S$, in contrast to our algorithms. Our algorithms can be applied to MAB to achieve the regret bound (Theorem 2) of
$$\tilde{O}\left(K\sqrt{\sum_{t=1}^{T} \|\ell_t - \bar{\ell}_t\|_\infty^2}\right)$$
for this special case, which is inferior to previous results of $\tilde{O}(\sqrt{V})$ and $\tilde{O}(\sqrt{KS})$ achieved by MAB-specialized algorithms. Algorithms based on continuous MWU, such as the one by Hazan and Karnin [32] and ours, may achieve worst-case optimal regret for general linear bandits, but, for MAB, they seem not competitive with MAB-specialized algorithms as they do not exploit specific structures of the action set.

# 3 Preliminaries

Given a vector $x \in \mathbb{R}^d$ and a positive-semidefinite matrix $M \in \mathbb{R}^{d \times d}$, let $\|x\|_M = \sqrt{x^\top M x}$. For symmetric matrices $A, B$, we denote $A \succeq B$ if $A - B$ is positive-semidefinite. We denote the convex hull of $\mathcal{A}$ by $\mathcal{A}'$. Given a distribution $p$ over $\mathcal{A}'$, define a vector $\mu(p) \in \mathbb{R}^d$ and a matrix $S(p) \in \mathbb{R}^{d \times d}$ by

$$\mu(p) = \mathop{\mathbf{E}}_{x \sim p}[x], \quad S(p) = \mathop{\mathbf{E}}_{x \sim p}[xx^\top]. \tag{2}$$

For ease of exposition, we also use $p$ to denote its density function simultaneously. If the density function $p : \mathbb{R}^d \to \mathbb{R}_{\geq 0}$ of a probability distribution has a convex support and $\log(p(x))$ is a concave function (on the support), then we call the distribution *log-concave*. We will make use of the following concentration property of log-concave distributions:

**Lemma 1.** *If $x$ follows a log-concave distribution $p$ over $\mathbb{R}^d$ and $S(p) \preceq I$, we have*

$$\Pr[\|x\|_2^2 \geq d\alpha^2] \leq d \exp(1 - \alpha) \tag{3}$$

*for arbitrary $\alpha \geq 0$.*

This lemma follows from, e.g., Lemma 5.7 in [42]. A complete proof can be found in Appendix A.

## 3.1 Adversarial Linear Bandits

In the adversarial linear bandit problem, a player is given an action set $\mathcal{A} \subseteq \mathbb{R}^d$, which is assumed to be a compact set in $\mathbb{R}^d$, before the game starts. Without loss of generality, we assume that $\mathcal{A}$ is not contained in any proper linear subspace. Note that the action set $\mathcal{A}$ can be discrete. In each round $t \in [T]$, the player chooses an action $a_t \in \mathcal{A}$, and then the environment reveals the loss $\langle \ell_t, a_t \rangle$, where the loss vector $\ell_t \in \mathbb{R}^d$ is in the convex set $\mathcal{L} \subseteq \mathbb{R}^d$ defined as

$$\mathcal{L} = \{\ell \in \mathbb{R}^d \mid -1 \leq \langle \ell, a \rangle \leq 1 \text{ for all } a \in \mathcal{A}\}, \tag{4}$$

and hence $\langle \ell_t, a_t \rangle \in [-1, 1]$ holds.

In Section 4.1, we consider a problem setting with *predicted loss vectors* $m_t \in \mathbb{R}^d$. In this problem setting, the player is given $m_t$ before choosing $a_t$. The predicted loss vectors $(m_t)$ are arbitrary sequences over $\mathbb{R}^d$, and each $m_t$ may be chosen depending on $\{(a_j, \ell_j)\}_{j < t}$.

When we discuss first-order regret bounds, we assume that the loss $\langle \ell_t, a \rangle$ is non-negative for any $a \in \mathcal{A}$. This assumption is a standard one when discussing first-order bounds (e.g., [6]) and is indispensable for ensuring that $\sum_{t=1}^T \langle \ell_t, a^* \rangle \geq 0$. When we discuss second-order regret bounds, we fix a norm $\| \cdot \|$ over $\mathbb{R}^d$ such that $\max_{a \in \mathcal{A}} \|a\| \leq 1$. Let $\| \cdot \|_*$ denote the dual norm of $\| \cdot \|$, i.e., $\|\ell\|_* = \max_{\|x\| \leq 1} \langle \ell, x \rangle$.

When we consider computational complexity, we assume that we can solve linear optimization over $\mathcal{A}$, i.e., there exists an oracle $\mathcal{O}$ with which given $\ell \in \mathbb{R}^d$, we can compute $\mathcal{O}(\ell) \in \operatorname{argmin}_{a \in \mathcal{A}} \{\langle \ell, a \rangle\}$. Such an assumption is standard in the context of online optimization [24; 33; 38] as this is an almost minimum assumption for developing computationally efficient online optimization algorithms with sublinear regret bounds.

# 4 Algorithms and Regret Upper Bounds

In this section, we explain our algorithms and analyze their regret bounds. In Section 4.1, we provide an algorithm (Algorithm 1) for a scenario in which predicted loss vectors $m_t$ are available, and we analyze its regret bound, which holds for arbitrary $m_t$. In Section 4.2, we show that Algorithm 1 enjoys a second-order regret bound when we choose $m_t$ in a sophisticated way on the basis of the observed feedback. In Section 4.3, this algorithm with a second-order bound is shown to have a first-order regret bound as well,

## 4.1 Algorithm with predicted loss vectors

Let us first consider here the case in which predicted loss vectors $m_t \in \mathbb{R}^d$ for $\ell_t$ are available. In this setting, the player is given $m_t$ before choosing $a_t$. We assume that $\langle m_t, a \rangle \in [-1, 1]$ holds for any $a \in \mathcal{A}$.

**Algorithm 1** An algorithm for adversarial linear bandits with predicted loss
1: **for** $t = 1, 2, \ldots, T$ **do**
2:     **repeat**
3:         Pick $x_t$ from the distribution $p_t$, defined by (5).
4:     **until** $\|x_t\|^2_{S(p_t)^{-1}} \leq d\gamma_t^2$
5:     Choose $a_t \in \mathcal{A}$ so that $\mathbf{E}[a_t] = x_t$, play $a_t$, and receive a loss $\langle \ell_t, a_t \rangle$ as feedback.
6:     Compute an unbiased estimator $\hat{\ell}_t$ of $\ell_t$ as $\hat{\ell}_t = m_t + \langle \ell_t - m_t, a_t \rangle \cdot S(\tilde{p}_t)^{-1} x_t$.
7:     Update $p_t$ as in (5).
8: **end for**

Our algorithm maintains probability distributions over $\mathcal{A}'$, the convex hull of $\mathcal{A}$, following the multiplicative weight update method [7].

$$w_t(x) = \exp\left(-\eta_t \left\langle \sum_{j=1}^{t-1} \hat{\ell}_j + m_t, x \right\rangle \right), \quad p_t(x) = \frac{w_t(x)}{\int_{y \in \mathcal{A}'} w_t(y) \mathrm{d}y} \quad (x \in \mathcal{A}'), \qquad (5)$$

where $\eta_j > 0$ are parameters referred to as *learning rates*, which we will determine later, and each $\hat{\ell}_j \in \mathbb{R}^d$ is an unbiased estimator of $\ell_j$ defined below. First, define the *truncated distribution* $\tilde{p}_t$ of $p_t$ as

$$\tilde{p}_t(x) = \frac{p_t(x)\mathbf{1}\{\|x\|^2_{S(p_t)^{-1}} \leq d\gamma_t^2\}}{\mathrm{Pr}_{y \sim p_t}[\|y\|^2_{S(p_t)^{-1}} \leq d\gamma_t^2]} \propto p_t(x)\mathbf{1}\{\|x\|^2_{S(p_t)^{-1}} \leq d\gamma_t^2\}, \qquad (6)$$

where $\gamma_t > 1$ is a parameter we will define later. In each round, the algorithm samples $x_t \in \mathcal{A}'$ according to $\tilde{p}_t$, and then chooses $a_t \in \mathcal{A}$ so that $\mathbf{E}[a_t | x_t] = x_t$. More precisely, we compute $\lambda_1, \ldots, \lambda_{d+1} \geq 0$ and $b_1, \ldots, b_{d+1} \in \mathcal{A}$ such that $\sum_{i=1}^{d+1} \lambda_i = 1$ and $\sum_{i=1}^{d+1} \lambda_i b_i = x_t$, and then output $a_t = b_i$ with probability $\lambda_i$. Such $\{(\lambda_i, b_i)\}_{i=1}^{d+1}$ can be efficiently computed given a linear optimization oracle for $\mathcal{A}$, e.g., via the ellipsoid method as stated in Corollary 14.1g in [46]. After taking the action $a_t$, the algorithm receives a loss $\langle \ell_t, a_t \rangle$ as feedback, and it constructs an unbiased estimator $\hat{\ell}_t$ of $\ell_t$ as follows:

$$\hat{\ell}_t = m_t + (\langle \ell_t, a_t \rangle - \langle m_t, a_t \rangle)S(\tilde{p}_t)^{-1} x_t. \qquad (7)$$

We note that $S(\tilde{p}_t)$ is invertible. This follows from the assumption that $\mathcal{A}$ is not contained in any proper linear subspace. Indeed, under this assumption, $\mathcal{A}'$ is a full-dimensional convex set with a positive Lebesgue measure. Combining this and Lemma 1, we can see that the domain of $\tilde{p}_t$ is full-dimensional as well. Therefore, the distribution $\tilde{p}_t$ has a density function taking positive values over a full-dimensional convex set, which implies that $S(\tilde{p}_t)$ is positive-definite. A similar argument can be found, e.g., in [36] (between Eq. (4) and (5)).

**Lemma 2.** *The vector $\hat{\ell}_t$ is an unbiased estimator of $\ell_t$, i.e., we have $\mathbf{E}[\hat{\ell}_t | \ell_t] = \ell_t$.*

*Proof.* The expectation of $\hat{\ell}_t$ is

$$\mathop{\mathbf{E}}_{x_t, a_t}[\hat{\ell}_t | \ell_t] = m_t + S(\tilde{p}_t)^{-1} \mathop{\mathbf{E}}_{x_t, a_t}[x_t a_t^\top](\ell_t - m_t). \qquad (8)$$

Since $\mathbf{E}[a_t | x_t] = x_t$, we have $\mathbf{E}_{x_t, a_t}[x_t a_t^\top] = \mathbf{E}_{x_t}[x_t x_t^\top] = S(\tilde{p}_t)$. Combining this and (8), we obtain $\mathbf{E}[\hat{\ell}_t | \ell_t] = \ell_t$. $\qquad \square$

Our algorithm can be summarized in Algorithm 1. Algorithm 1 enjoys the following regret bound:

**Theorem 1.** *Suppose $\gamma_t \geq 4\log(10dt)$ and $\eta_t \leq \frac{1}{\sqrt{800d\gamma_t}}$ for all $t$. Then, for any $a^* \in \mathcal{A}$, Algorithm 1 satisfies*

$$\mathbf{E}[R_T(a^*)] \leq d \cdot \mathbf{E}\left[4 \sum_{t=1}^{T} \eta_t \gamma_t^2 (\langle \ell_t - m_t, a_t \rangle)^2 + \frac{\log T}{\eta_T}\right] + 3. \qquad (9)$$

*Consequently, by setting* $\gamma_t = 4\log(10dt)$ *and* $\eta_t = \left(800d\gamma_t^2 + 16\sum_{j=1}^{t-1}\gamma_j^2(\langle\ell_j - m_j, a_j\rangle)^2\right)^{-\frac{1}{2}}$, *we obtain*

$$\mathbf{E}[R_T(a^*)] \le 32d\log T \cdot \log(10dT) \cdot \mathbf{E}\left[\sqrt{\sum_{t=1}^{T}(\langle\ell_t - m_t, a_t\rangle)^2 + 50d}\right]. \tag{10}$$

This regret bound can be shown via analyses for the optimistic follow-the-regularized-leader algorithm [45; 44] and the continuous multiplicative weight update method [7; 32], combined with Lemmas 1 and 2. A complete proof is given in Section B in the appendix.

## 4.2 Second-order regret bound

In this section, we show that we can obtain a second-order bound by appropriately choosing $m_t$ in Algorithm 1, by means of an online learning technique. Consider $m_t$ defined by

$$m_t \in \underset{m\in\mathcal{L}}{\arg\min}\left\{\|m\|_S^2 + \sum_{j=1}^{t-1}(\langle\ell_j - m, a_j\rangle)^2\right\}, \tag{11}$$

where $\mathcal{L} \subseteq \mathbb{R}^d$ is defined as in (4) and $S \in \mathbb{R}^{d\times d}$ is an arbitrary positive-definite matrix.

**Lemma 3.** *If $m_t$ is given by (11), we have, for any $m^* \in \mathcal{L}$,*

$$\sum_{t=1}^{T}(\langle\ell_t - m_t, a_t\rangle)^2 \le \sum_{t=1}^{T}(\langle\ell_t - m^*, a_t\rangle)^2 + \|m^*\|_S^2 + 16d\log\left(1 + \frac{T}{d}\max_{a\in\mathcal{A}}\|a\|_{S^{-1}}^2\right). \tag{12}$$

This lemma can be shown by following the analysis of *online ridge regression*, e.g., see the proof of Theorem 11.7 in [17]. To further bound the right-hand side of (12), we provide a specific example of $S$. We first note that there exists a matrix $S \in \mathbb{R}^{d\times d}$ such that

$$\|m\|_S^2 \le d \text{ for any } m \in \mathcal{L}, \quad \|a\|_{S^{-1}}^2 \le 4d \text{ for any } a \in \mathcal{A}, \tag{13}$$

and given a linear optimization oracle over $\mathcal{A}$, one can compute such an $S$ efficiently, via a *barycentric spanner* [12] for $\mathcal{A}$. More precise method for constructing $S$ is described in Section C in the appendix.

Combining Theorem 1, Lemma 3 and (13), we obtain the following regret bound:

**Theorem 2.** *Suppose that $\gamma_t$, $\eta_t$, and $m_t$ are given by (32), (11) (with $\mathcal{L}$ as in (4)), and (34), respectively. Then the actions $a_t$ of Algorithm 1 satisfy*

$$\mathbf{E}[R_T(a^*)] \le 32d\log T \cdot \log(10dT) \cdot \mathbf{E}\left[\sqrt{\sum_{t=1}^{T}(\langle\ell_t - m^*, a_t\rangle)^2 + 51d + 16d\log(1 + 4T)}\right]$$

$$\le 32d\log T\log(10dT) \cdot \mathbf{E}\left[\sqrt{\sum_{t=1}^{T}\|\ell_t - m^*\|_*^2 + 51d + 16d\log(1 + 4T)}\right].$$

*for arbitrary $a^* \in \mathcal{A}$ and $m^* \in \mathcal{L}$.*

## 4.3 First-order regret bound

In this section, we show that the bound in the Theorem 2 can be used to obtain a first-order regret bound assuming that $0 \le \langle\ell_t, a_t\rangle \le 1$.

**Theorem 3.** *Suppose that the assumptions in Theorem 2 hold and that the observed losses are non-negative. Then we have, for $\xi = O(\log d \cdot \log^2 T)$,*

$$\mathbf{E}[R_T(a^*)] = O\left(\xi d\sqrt{\mathbf{E}\left[\sum_{t=1}^{T}\langle\ell_t, a^*\rangle\right]} + \xi^2 d^2\right). \tag{14}$$

*Proof.* For notational simplicity, let $C_1 = 32d \log T \cdot \log(10dT)$ and $C_2 = 51d + 16d \log(1 + 4T)$. From the inequality in Theorem 2 with $m^* = 0$, we have

$$\mathbf{E}[R_T(a^*)] \leq C_1 \mathbf{E}\left[\sqrt{\sum_{t=1}^{T}(\langle \ell_t, a_t \rangle)^2 + C_2}\right] \leq C_1 \mathbf{E}\left[\sqrt{\sum_{t=1}^{T}\langle \ell_t, a_t \rangle + C_2}\right]$$

$$= C_1 \mathbf{E}\left[\sqrt{\sum_{t=1}^{T}\langle \ell_t, a^* \rangle + R_T(a^*) + C_2}\right] \leq C_1 \sqrt{\mathbf{E}\left[\sum_{t=1}^{T}\langle \ell_t, a^* \rangle\right] + \mathbf{E}[R_T(a^*)] + C_2},$$

where the second inequality follows from the assumption of $0 \leq \langle \ell_t, a_t \rangle \leq 1$, the equality follows from the definition (1) of $R_T$, and the last inequality follows from Jensen's inequality. By solving the quadratic inequality with respect to $\mathbf{E}[R_T(a^*)]$, we obtain $\mathbf{E}[R_T(a^*)] \leq C_1\left(\frac{C_1}{2} + \frac{1}{2}\sqrt{C_1^2 + 4\left(C_2 + \mathbf{E}\left[\sum_{t=1}^{T}\langle \ell_t, a^* \rangle\right]\right)}\right) \leq C_1\sqrt{C_1^2 + 2\left(C_2 + \mathbf{E}\left[\sum_{t=1}^{T}\langle \ell_t, a^* \rangle\right]\right)}$, where the last inequality follows from $\frac{\sqrt{x}+\sqrt{y}}{2} \leq \sqrt{\frac{x+y}{2}}$ for $x, y \geq 0$. $\square$

*Remark* 1. The regret bound in Theorem 3 holds even when $m_t$ is chosen to be $m_t = 0$ for all $t$, as can be seen in the proof. However, we would like to stress here that, by setting $m_t$ as in Theorem 2, a *single* algorithm *simultaneously* enjoys two different regret bounds as in Theorems 2 and 3.

### 4.4 Computationally efficient implementation

Algorithm 1 can be implemented in a computationally efficient way, assuming that linear optimization over $\mathcal{A}$ can be efficiently solved. As shown in [42], we can get a sample from a log-concave distribution $p_t$ in a polynomial time in $d$, if we can compute $w_t(x) \propto p_t(x)$ and can access a membership oracle for $\mathrm{supp}(p_t) = \mathcal{A}'$, i.e., we can decide whether a given vector $x \in \mathbb{R}^d$ belongs to $\mathcal{A}'$. The membership problem can be reduced to linear optimization problems by ellipsoid methods, as shown, e.g., in Corollary 14.1b in [46]. Consequently, we can get a sample from $p_t$ in polynomial time. The matrix $S(\tilde{p}_t)$ (7) can be efficiently computed as well. Indeed, since $\tilde{p}_t$ is log-concave, for any $\varepsilon > 0$, we can get an $\varepsilon$-approximation of $S(\tilde{p}_t)$ w.h.p. by generating $(d/\varepsilon)^{O(1)}$ samples from $\tilde{p}_t$, from Corollary 2.7 of [42]. Samples from $\tilde{p}_t$ can be generated with their polynomial-time sampling algorithm as mentioned in Section 4.4 of our manuscript. A similar discussion can be found in Lemma 5.17 of [14].

The vector $m_t$ defined in (11) can be computed efficiently as well. In fact, a linear optimization oracle for $\mathcal{A}$ immediately leads to a separation oracle for $\mathcal{L}$ defined by (4). Hence, we can solve a convex optimization over $\mathcal{L}$ such as (11), e.g., by using ellipsoid methods.

## 5 Lower Bound

In this section, we provide instance-dependent regret lower bounds. In what follows, we assume $d \geq 2$ and $\mathcal{A} \subseteq \{-1, 1\}^d$. Note that we then have $\|a\|_\infty \leq 1$ for any $a \in \mathcal{A}$.

**Theorem 4.** *Let $\mathcal{A} = \{-1, 1\}^{d-1} \times \{1\}$. For any algorithm and for any $L$ with $d^2 \leq L \leq T$, there exists a sequence $(\ell_t)_{t=1}^{T}$ of $d$-dimensional loss vectors such that the following hold: (i) $0 \leq \langle \ell_t, a \rangle \leq 1$ for any $a \in \mathcal{A}$, (ii) $\min_{a^* \in \mathcal{A}} \sum_{t=1}^{T}\langle \ell_t, a^* \rangle \leq L$, (iii) $\sum_{t=1}^{T}\|\ell_t\|_1^2 \leq L$, and (iv) any algorithm satisfies $\max_{a^* \in \mathcal{A}} \mathbf{E}[R_T(a^*)] = \Omega(d\sqrt{L})$.*

This theorem complements Theorems 1, 2 and 3 by providing almost matching lower bounds. Indeed, for any problem instances satisfying (iii) with $L \geq d^2$, both Theorem 1 (with $m_t = 0$) and Theorem 2 imply $\max_{a^* \in \mathcal{A}} \mathbf{E}[R_T(a^*)] = \tilde{O}(d\sqrt{L})$, which matches the lower bound of (iv) in Theorem 4. Similarly, for any problem instances satisfying (i) and (ii) with $L \geq d^2$, Theorem 3 implies $\max_{a^* \in \mathcal{A}} \mathbf{E}[R_T(a^*)] = \tilde{O}(d\sqrt{L})$.

Theorem 4 can be shown by adopting the hard instances used to show the worst-case lower bound of $\Omega(d\sqrt{T})$, e.g., by Dani et al. [25]. The proof of Theorem 4 is given in Appendix D.

## 6   Conclusion

In this paper, we provided algorithms with nearly tight first- and second-order regret bounds for adversarial linear bandit problems, with the aid of techniques such as optimistic online optimization and properties of log-concave distributions. A future research direction is to obtain improved path-length regret bounds, as discussed, e.g., in [16; 48]. Another direction would be to improve practical computational efficiency. Our proposed algorithms require large computational time due to the complexity of continuous multiplicative weight update, though it is of polynomial in dimensions. Hence, algorithms by Hazan and Kale [30]; Rakhlin and Sridharan [44] have smaller runtime if the action set admits a self-concordant barrier that can be computed efficiently.

## Broader Impact

This is a theoretical work and does not present any foreseeable societal consequences.

## Acknowledgments and Disclosure of Funding

SI was supported by JST, ACT-I, Grant Number JPMJPR18U5, Japan. SH was supported by JST, ACT-I, Grant Number JPMJPR17UM, Japan. TS was supported by JST, ERATO, Grant Number JPMJER1903, Japan. YY was supported by JSPS KAKENHI Grant Number 18H05291, Japan.

## Footnotes

[2]In fact, MAB is equivalent to the linear bandit problem with an action set being the standard basis of a $K$-dimensional space $\mathcal{A} = \{e_1, \ldots, e_K\} \subseteq \{0, 1\}^K$: Each element $\ell_{ti}$ of the loss vector $\ell_t \in [0, 1]^K$ stands for the loss incurred by choosing the $i$-th action $e_i$.

[3]The term "second-order regret" has been used to mean various bounds in the literature. In this work, we adopt the one stated here.

[4]In literature [19; 29], values $Q^*$ and $V^*$ were originally defined as $Q^* = \max_{\tau \in [T]} \sum_{t=1}^{\tau} \ell_{ta_\tau^*}^2$ and $V^* = \max_{\tau \in [T]} \sum_{t=1}^{\tau} (\ell_{ta_\tau^*} - \bar{\ell}_{\tau a_\tau^*})^2$ with $a_\tau^* \in \arg\min_{a \in [K]} \sum_{t=1}^{\tau} \ell_{ta}$ and $\bar{\ell}_\tau = \frac{1}{\tau} \sum_{t=1}^{\tau} \ell_t$.

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
