[Supplementary Material]

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

## A Proof of Lemma 1

*Proof.* Since a linear transformation of a log-concave random variable follows a log-concave distribution as well (Theorem 5.1 in [42]), each $x_i$ follows a log-concave distribution and we have $\mathbf{E}[x_i^2] \leq 1$ from the assumption of $S(p) \preceq I$. Then, we have

$$\Pr[\|x\|_2^2 \geq d\alpha^2] \leq \Pr[\exists i \in [d], x_i^2 \geq \alpha^2] \leq \sum_{i=1}^d \Pr[|x_i| \geq \alpha] \leq d \exp(1 - \alpha), \quad (15)$$

where the last inequality follows from Lemma 5.7 in [42]. $\qquad\square$

## B Proof of Theorem 1

Here, we provide a proof of this Theorem 1, and hereafter, we assume that $\gamma_t$ and $\ell_t$ satisfy the assumptions in Theorem 1.

Since we have $\mathbf{E}[a_t|\tilde{p}_t] = \mathbf{E}[x_t|\tilde{p}_t] = \mu(\tilde{p}_t)$, the expected regret can be expressed as

$$\mathbf{E}[R_T(a^*)] = \mathbf{E}\left[\sum_{t=1}^T \langle \ell_t, a_t - a^* \rangle\right] = \mathbf{E}\left[\sum_{t=1}^T \langle \ell_t, \mu(\tilde{p}_t) - a^* \rangle\right]$$

$$= \mathbf{E}\left[\sum_{t=1}^T \langle \ell_t, \mu(\tilde{p}_t) - \mu(p_t) \rangle\right] + \mathbf{E}\left[\sum_{t=1}^T \langle \hat{\ell}_t, \mu(p_t) - a^* \rangle\right] \quad (16)$$

where the last equality follows from Lemma 2. The first term in (16) can be bounded by using Lemma 1. Indeed, we can show that $\tilde{p}_t$ is *close* to $p_t$ using the following lemma:

**Lemma 4.** *For any function $f : \mathcal{A}' \to [-1, 1]$ we have*

$$\left| \mathbf{E}_{x \sim p_t}[f(x)] - \mathbf{E}_{x \sim \tilde{p}_t}[f(x)] \right| \leq 10d \exp(-\gamma_t) \leq \frac{1}{2t^2}. \quad (17)$$

*Further, we have*

$$\frac{3}{4}S(p_t) \preceq S(\tilde{p}_t) \preceq \frac{4}{3}S(p_t) \quad (18)$$

*Proof.* From the definition (6) of $\tilde{p}_t$, we have

$$\mathbf{E}_{x \sim \tilde{p}_t}[f(x)] = \frac{1}{\Pr_{x \sim p_t}[\|x\|_{S(p_t)^{-1}}^2 \leq d\gamma_t^2]} \int_{x \in \mathcal{A}'} f(x)\mathbf{1}\{\|x\|_{S(p_t)^{-1}}^2 \leq d\gamma_t^2\}p_t(x)\mathrm{d}x$$

$$= \frac{1}{1-\delta} \int_{x \in \mathcal{A}'} f(x)\mathbf{1}\{\|x\|_{S(p_t)^{-1}}^2 \leq d\gamma_t^2\}p_t(x)\mathrm{d}x$$

$$= \frac{1}{1-\delta} \left( \mathbf{E}_{x \sim p_t}[f(x)] - \int_{x \in \mathcal{A}'} f(x)\mathbf{1}\{\|x\|_{S(p_t)^{-1}}^2 > d\gamma_t^2\}p_t(x)\mathrm{d}x \right),$$

where we denote $\delta = \Pr_{x \sim p_t}[\|x\|_{S(p_t)^{-1}}^2 > d\gamma_t^2]$. From this expression, we have

$$\left| \mathbf{E}_{x \sim p_t}[f(x)] - \mathbf{E}_{x \sim \tilde{p}_t}[f(x)] \right| = \frac{1}{1-\delta} \left| \delta \mathbf{E}_{x \sim p_t}[f(x)] + \int_{x \in \mathcal{A}'} f(x)\mathbf{1}\{\|x\|_{S(p_t)^{-1}}^2 > d\gamma_t^2\}p_t(x)\mathrm{d}x \right|$$

$$\leq \frac{1}{1-\delta} \left( \delta \mathbf{E}_{x \sim p_t}[1] + \int_{x \in \mathcal{A}'} \mathbf{1}\{\|x\|_{S(p_t)^{-1}}^2 > d\gamma_t^2\}p_t(x)\mathrm{d}x \right) = \frac{2\delta}{1-\delta}, \quad (19)$$

where the inequality follows from the assumption that $f(x) \in [-1, 1]$. Since $p_t$ is a log-concave distribution, we can apply Lemma 1 to $x = S(p_t)^{-1/2}y$ with $y \sim p_t$. In fact, assumptions in Lemma 1 hold since we have $\mathbf{E}[xx^\top] = S(p_t)^{-1/2}\mathbf{E}[yy^\top]S(p_t)^{-1/2} = S(p_t)^{-1/2}S(p_t)S(p_t)^{-1/2} = I$ and

since log-concavity is preserved under any liner transformation. Using Lemma 1 for $x = S(p_t)^{-1/2}y$, we have

$$\delta = \Pr_{x \sim p_t}[\|x\|^2_{S(p_t)^{-1}} > d\gamma_t^2] \le d\exp(1 - \gamma_t) \le 3d\exp(-\gamma_t) \le \frac{1}{6t^2}, \tag{20}$$

where the last inequality follows from $\gamma_t \ge 4\log(10dt)$. Combining (19) and (20), we obtain (17). We next show (18). For any $y \in \mathbb{R}^d$, we have

$$y^\top S(\tilde{p}_t)y = \mathbf{E}_{x \sim \tilde{p}_t}\left[(y^\top x)^2\right] = \frac{1}{1-\delta}\mathbf{E}_{x \sim p_t}\left[(y^\top x)^2 \mathbf{1}\{\|x\|^2_{S(p_t)^{-1}} \le d\gamma_t^2\}\right]$$

$$\le \frac{1}{1-\delta}\mathbf{E}_{x \sim p_t}\left[(y^\top x)^2\right] = \frac{1}{1-\delta}y^\top S(p)y.$$

Since this holds for all $y \in \mathbb{R}^d$ and $\frac{1}{1-\delta} \le 4/3$, the second inequality in (18) holds. Furthermore, we have

$$y^\top S(p_t)y - y^\top S(\tilde{p}_t)y = \mathbf{E}_{x \sim p_t}\left[(y^\top x)^2\right] - \frac{1}{1-\delta}\mathbf{E}_{x \sim p_t}\left[(y^\top x)^2 \mathbf{1}\{\|x\|^2_{S(p_t)^{-1}} \le d\gamma_t^2\}\right]$$

$$\le \mathbf{E}_{x \sim p_t}\left[(y^\top x)^2 \mathbf{1}\{\|x\|^2_{S(p_t)^{-1}} > d\gamma_t^2\}\right]$$

$$\le y^\top S(p_t)y \mathbf{E}_{x \sim p_t}\left[\|x\|^2_{S(p_t)^{-1}} \mathbf{1}\{\|x\|^2_{S(p_t)^{-1}} > d\gamma_t^2\}\right], \tag{21}$$

where the last inequality follows from the Cauchy–Schwarz inequality:

$$(y^\top x)^2 = \left(\left\langle S(p_t)^{\frac{1}{2}}y, S(p_t)^{-\frac{1}{2}}x\right\rangle\right)^2 \le \|S(p_t)^{\frac{1}{2}}y\|_2^2 \cdot \|S(p_t)^{-\frac{1}{2}}x\|_2^2 = y^\top S(p_t)y \cdot \|x\|^2_{S(p_t)^{-1}}.$$

The right-hand side of (21) can be bounded by using Lemma 1 as follows:

$$\mathbf{E}_{x \sim p_t}\left[\|x\|^2_{S(p_t)^{-1}} \mathbf{1}\{\|x\|^2_{S(p_t)^{-1}} > d\gamma_t^2\}\right]$$

$$\le \sum_{n=1}^{\infty}(n+1)^2 d\gamma_t^2 \Pr_{x \sim p_t}\left[n^2 d\gamma_t^2 \le \|x\|^2_{S(p)^{-1}} \le (n+1)^2 d\gamma_t^2\right]$$

$$\le \sum_{n=1}^{\infty}(n+1)^2 d\gamma_t^2 \cdot d\exp(1 - n\gamma_t)$$

$$\le d^2\gamma_t^2 \sum_{n=1}^{\infty}\exp(2 + n - n\gamma_t) = d^2\gamma_t^2 \frac{\exp(3 - \gamma_t)}{1 - \exp(1 - \gamma_t)} \le \frac{1}{4} \tag{22}$$

where the second inequality follows from Lemma 1, the second inequality comes from $y^2 \le \exp(y)$ for $y \le 0$, and the last inequality follows from the assumption of $\gamma_t \ge 4\log(10dt)$. Combining (21), (22) and the assumption of $\gamma_t \ge 4\log(10dt)$, we obtain the first inequality of (18). $\square$

Since $\langle \ell_t, x \rangle \in [-1, 1]$ for all $x \in \mathcal{A}'$, (17) implies $|\langle \ell_t, \mu(\tilde{p}_t) - \mu(p_t)\rangle| \le 1/(2t^2)$.

The second term in (16) can be bounded by following the analysis of optimistic mirror descent [45]:

**Lemma 5.** *For any $a^* \in \mathcal{A}$, we have*

$$\sum_{t=1}^{T}\left\langle \hat{\ell}_t, \mu(p_t) - a^* \right\rangle \le \sum_{t=1}^{T}\left(\frac{1}{\eta_t}\mathbf{E}_{x \sim p_t}\left[\psi\left(-\eta_t\left\langle \hat{\ell}_t - m_t, x \right\rangle\right)\right]\right) + \frac{d\log T}{\eta_T} + \frac{1}{T}\sum_{t=1}^{T}\left\langle \hat{\ell}_t, \bar{a} - a^* \right\rangle,$$

*where*

$$\psi(y) = \exp(y) - y - 1, \quad \bar{a} = \mu(p_0). \tag{23}$$

*Proof.* Define $v_t : \mathcal{A}' \to \mathbb{R}$ and $u_t : \mathcal{A}' \to \mathbb{R}$ by

$$u_t(x) = \exp\left(-\eta_t\left\langle \sum_{j=1}^{t}\hat{\ell}_j, x \right\rangle\right), \quad v_t(x) = \exp\left(-\eta_{t+1}\left\langle \sum_{j=1}^{t}\hat{\ell}_j, x \right\rangle\right), \tag{24}$$

and define

$$U_t = \int_{x \in \mathcal{A}'} u_t(x)\mathrm{d}x, \quad V_t = \int_{x \in \mathcal{A}'} v_t(x)\mathrm{d}x, \quad W_t = \int_{x \in \mathcal{A}'} w_t(x)\mathrm{d}x. \tag{25}$$

Since $u_t$ can be expressed as $u_t(x) = w_t(x)\exp\left(-\eta_t \left\langle \hat{\ell}_t - m_t, x \right\rangle\right)$ from the definitions (5) and (24) of $w_t$ and $u_t$, respectively, we have

$$U_t = \int_{x \in \mathcal{A}'} w_t(x)\exp\left(-\eta_t \left\langle \hat{\ell}_t - m_t, x \right\rangle\right)\mathrm{d}x = W_t \cdot \mathop{\mathbf{E}}_{x \sim p_t}\left[\exp\left(-\eta_t \left\langle \hat{\ell}_t - m_t, x \right\rangle\right)\right]$$

$$= W_t \cdot \left(1 - \eta_t \left\langle \hat{\ell}_t - m_t, \mu(p_t) \right\rangle + \mathop{\mathbf{E}}_{x \sim p_t}\left[\psi\left(-\eta_t \left\langle \hat{\ell}_t - m_t, x \right\rangle\right)\right]\right).$$

By taking the logarithms of both sides, we obtain

$$\log U_t = \log W_t + \log\left(1 - \eta_t \left\langle \hat{\ell}_t - m_t, \mu(p_t) \right\rangle + \mathop{\mathbf{E}}_{x \sim p_t}\left[\psi\left(-\eta_t \left\langle \hat{\ell}_t - m_t, x \right\rangle\right)\right]\right)$$

$$\leq \log W_t - \eta_t \left\langle \hat{\ell}_t - m_t, \mu(p_t) \right\rangle + \mathop{\mathbf{E}}_{x \sim p_t}\left[\psi\left(-\eta_t \left\langle \hat{\ell}_t - m_t, x \right\rangle\right)\right],$$

where we used the inequality $\log(1 + x) \leq x$ for $x > -1$. The condition $x > -1$ indeed holds since $x$ here corresponds to $x = -1 + \mathbf{E}[\exp(-\eta_t \langle \hat{\ell}_t - m_t, x \rangle)]$. Hence, we have

$$\left\langle \hat{\ell}_t - m_t, \mu(p_t) \right\rangle \leq \frac{1}{\eta_t}\left(\log \frac{W_t}{U_t} + \mathop{\mathbf{E}}_{x \sim p_t}\left[\psi\left(-\eta_t \left\langle \hat{\ell}_t - m_t, x \right\rangle\right)\right]\right). \tag{26}$$

Similarly, since we have

$$V_{t-1} = \int_{x \in \mathcal{A}'} w_t(x)\exp\left(\eta_t \left\langle m_t, x \right\rangle\right)\mathrm{d}x = W_t \cdot \mathop{\mathbf{E}}_{x \sim p_t}\left[\exp\left(\eta_t \left\langle m_t, x \right\rangle\right)\right] \geq W_t \cdot \exp(\eta_t \left\langle m_t, \mu(p_t) \right\rangle),$$

where we applied Jensen's inequality, it holds that

$$\left\langle m_t, \mu(p_t) \right\rangle \leq \frac{1}{\eta_t}\log\frac{V_{t-1}}{W_t}. \tag{27}$$

Combining (26) and (27) and taking the sum over $t = 1, \ldots, T$, we obtain

$$\sum_{t=1}^{T}\left\langle \hat{\ell}_t, \mu(p_t) \right\rangle \leq \sum_{t=1}^{T}\frac{1}{\eta_t}\left(\log\frac{V_{t-1}}{U_t} + \mathop{\mathbf{E}}_{x \sim p_t}\left[\psi\left(-\eta_t \left\langle \hat{\ell}_t - m_t, x \right\rangle\right)\right]\right). \tag{28}$$

Furthermore, noting that $V_0 = U_0 = \mathrm{vol}(\mathcal{A}')$, we have

$$\sum_{t=1}^{T}\frac{1}{\eta_t}\log\frac{V_{t-1}}{U_t} = \sum_{t=1}^{T}\frac{1}{\eta_t}\left(\log\frac{V_{t-1}}{V_0} - \log\frac{U_t}{U_0}\right)$$

$$= \sum_{t=1}^{T-1}\left(\frac{1}{\eta_{t+1}}\log\frac{V_t}{V_0} - \frac{1}{\eta_t}\log\frac{U_t}{U_0}\right) - \frac{1}{\eta_T}\log\frac{U_T}{U_0} \leq -\frac{1}{\eta_T}\log\frac{U_T}{U_0}, \tag{29}$$

where the inequality follows from the assumption of $\eta_{t+1} \leq \eta_t$ and Jensen's inequality, as follows:

$$\frac{1}{\eta_{t+1}}\log\frac{V_t}{V_0} = \frac{1}{\eta_{t+1}}\log\mathop{\mathbf{E}}_{x \sim p_0}\left[\exp\left(-\eta_{t+1}\left\langle\sum_{j=1}^{t}\hat{\ell}_j, x\right\rangle\right)\right]$$

$$= \frac{1}{\eta_{t+1}}\log\mathop{\mathbf{E}}_{x \sim p_0}\left[\exp\left(-\eta_t\left\langle\sum_{j=1}^{t}\hat{\ell}_j, x\right\rangle\right)^{\frac{\eta_{t+1}}{\eta_t}}\right]$$

$$\leq \frac{1}{\eta_{t+1}}\log\mathop{\mathbf{E}}_{x \sim p_0}\left[\exp\left(-\eta_t\left\langle\sum_{j=1}^{t}\hat{\ell}_j, x\right\rangle\right)\right]^{\frac{\eta_{t+1}}{\eta_t}}$$

$$= \frac{1}{\eta_t}\log\mathop{\mathbf{E}}_{x \sim p_0}\left[\exp\left(-\eta_t\left\langle\sum_{j=1}^{t}\hat{\ell}_j, x\right\rangle\right)\right] = \frac{1}{\eta_t}\log\frac{U_t}{U_0},$$

where the first and the last equalities follow from the definitions (24) and (25) of $U_t$ and $V_t$, and the inequality holds since the function $x \mapsto x^{\frac{\eta_{t+1}}{\eta_t}}$ $(x > 0)$ is a concave function. Set $\mathcal{A}_{a^*} := \{(1 - \frac{1}{T})a^* + \frac{1}{T}y \mid y \in \mathcal{A}\} \subseteq \mathcal{A}'$ and let $p^*$ denote a uniform distribution over $\mathcal{A}_{a^*}$. We then have

$$U_T \geq \int_{x \in \mathcal{A}_{a^*}} \exp\left(-\eta_T \left\langle \sum_{t=1}^{T} \hat{\ell}_t, x \right\rangle\right) \mathrm{d}x$$

$$= T^{-d} \int_{y \in \mathcal{A}'} \exp\left(-\eta_T \left\langle \sum_{t=1}^{T} \hat{\ell}_t, \left(1 - \frac{1}{T}\right)a^* + \frac{1}{T}y \right\rangle\right) \mathrm{d}y$$

$$\geq T^{-d} U_0 \exp\left(-\eta_T \left\langle \sum_{t=1}^{T} \hat{\ell}_t, \left(1 - \frac{1}{T}\right)a^* + \frac{1}{T}\bar{a} \right\rangle\right),$$

where $\bar{a} = \mu(p_0)$ and the last inequality follows from Jensen's inequality. Taking the logarithms of both sides, we have

$$-\frac{1}{\eta_T} \log \frac{U_T}{U_0} \leq \left\langle \sum_{t=1}^{T} \hat{\ell}_t, \left(1 - \frac{1}{T}\right)a^* + \frac{1}{T}\bar{a} \right\rangle + \frac{d \log T}{\eta_T} = \left\langle \sum_{t=1}^{T} \hat{\ell}_t, a^* + \frac{1}{T}(\bar{a} - a^*) \right\rangle + \frac{d \log T}{\eta_T}.$$

Combining this, (28), and (29), we obtain the desired inequality in the statement of Lemma 5. $\qquad\square$

We next evaluate the term $\mathbf{E}_{x \sim p_t}\left[\psi\left(-\eta_t \left\langle \hat{\ell}_t - m_t, x \right\rangle\right)\right]$ in Lemma 5. From the definition of $\psi$, $\psi(y) \leq y^2$ for $|y| \leq 1$, and hence $\mathbf{E}[\psi(y)]$ can be well approximated with $\mathbf{E}[y^2]$ if $y$ follows a log-concave distribution and $\mathbf{E}[y^2]$ is small enough:

**Lemma 6.** *If $y$ follows a log-concave distribution over $\mathbb{R}$ and if $\mathbf{E}[y^2] \leq 1/100$, we have*

$$\mathbf{E}[\psi(y)] \leq \mathbf{E}[y^2] + 30 \exp\left(-\frac{1}{\sqrt{\mathbf{E}[y^2]}}\right) \leq 2 \mathbf{E}[y^2] \quad where \quad \psi(x) = \exp(x) - x - 1. \quad (30)$$

*Proof.* Let $s = \sqrt{\mathbf{E}[x^2]}$. We have $s \leq 1/10$ from the assumption. We can bound $\mathbf{E}[\psi(x)]$ as follows:

$$\mathbf{E}[\psi(x)] = \mathbf{E}[\psi(x)\mathbf{1}\{x \leq 1\}] + \mathbf{E}[\psi(x)\mathbf{1}\{x > 1\}] \leq \mathbf{E}[x^2\mathbf{1}\{x \leq 1\}] + \mathbf{E}[\exp(x)\mathbf{1}\{x > 1\}]$$

$$\leq s^2 + \sum_{n=1}^{\infty} \exp(n+1) \Pr[n < x \leq n+1] \leq s^2 + \sum_{n=1}^{\infty} \exp(n+1) \Pr\left[\frac{n}{s} < \frac{x}{s}\right]$$

$$\leq s^2 + \sum_{n=1}^{\infty} \exp(n+1) \exp\left(1 - \frac{n}{s}\right) = s^2 + \frac{\exp(3 - s^{-1})}{1 - \exp(1 - s^{-1})} \leq s^2 + 30 \exp(-s^{-1}),$$

where the first inequality follows from $\psi(x) \leq x^2$ for $x \leq 1$ and $\psi(x) \leq \exp(x)$ for $x > 1$, the forth inequality follows from Lemma 1, and the last inequality follows from $s \leq 1/10$. By combining this and the fact that $30 \exp(-s^{-1}) \leq s^2$ for $0 < s \leq 1/10$, we obtain Lemma 6. $\qquad\square$

We give a bound for $\mathbf{E}_{x \sim p_t}\left[\psi\left(-\eta_t \left\langle \hat{\ell}_t - m_t, x \right\rangle\right)\right]$ by applying Lemma 6 with $y = -\eta_t \left\langle \hat{\ell}_t - m_t, x \right\rangle$. We can confirm that $y$ satisfies the assumption of Lemma 6 thanks to the truncated distribution that ensures $\|x_t\|_{S(p_t)^{-1}}^2 \leq d\gamma_t^2$. In fact, we have

$$\mathbf{E}_{x \sim p_t}\left[\left(-\eta_t \left\langle \hat{\ell}_t - m_t, x \right\rangle\right)^2\right] = \eta_t^2 (\hat{\ell}_t - m_t)^\top S(p_t)(\hat{\ell}_t - m_t)$$

$$= \eta_t^2 (\langle \hat{\ell}_t - m_t, a_t \rangle)^2 x_t^\top S(\tilde{p}_t)^{-1} S(p_t) S(\tilde{p}_t)^{-1} x_t \leq \frac{4}{3}\eta_t^2 (\langle \hat{\ell}_t - m_t, a_t \rangle)^2 x_t^\top S(\tilde{p}_t)^{-1} S(\tilde{p}_t) S(\tilde{p}_t)^{-1} x_t$$

$$= \frac{4}{3}\eta_t^2 (\langle \hat{\ell}_t - m_t, a_t \rangle)^2 \|x_t\|_{S(\tilde{p}_t)^{-1}}^2 \leq 2\eta_t^2 (\langle \hat{\ell}_t - m_t, a_t \rangle)^2 \|x_t\|_{S(p_t)^{-1}}^2 \leq 2d\gamma_t^2 \eta_t^2 (\langle \hat{\ell}_t - m_t, a_t \rangle)^2 \leq 1/100,$$

where the first and second inequalities follow from (18), the third inequality follows from $x_t$ being sampled from $\tilde{p}_t$ and the definition (6) of $\tilde{p}_t$, and the last inequality follows from the assumption of $\eta_t \leq \frac{1}{\sqrt{800d\gamma_t}}$. Hence, by Lemma 6, we have

$$\mathop{\mathbf{E}}_{x \sim p_t}\left[\psi\left(-\eta_t\left\langle \hat{\ell}_t - m_t, x\right\rangle\right)\right] \leq 4d\gamma_t^2\eta_t^2(\langle \ell_t - m_t, a_t\rangle)^2. \tag{31}$$

Combining this, (16), (17) and Lemma 5, we obtain

$$\mathbf{E}[R_T(a^*)] \leq \sum_{t=1}^T \frac{1}{2t^2} + \mathbf{E}\left[\sum_{t=1}^T 4d\gamma_t^2\eta_t(\langle \ell_t - m_t, a_t\rangle)^2 + \frac{d\log T}{\eta_T} + \frac{1}{T}\sum_{t=1}^T\left\langle \hat{\ell}_t, \bar{a} - a^*\right\rangle\right].$$

Since we have $\sum_{t=1}^T \frac{1}{t^2} \leq 2$ and since Lemma 2 implies that $\mathbf{E}\left[\left\langle \hat{\ell}_t, \bar{a} - a^*\right\rangle\right] = \langle \ell_t, \bar{a} - a^*\rangle \leq 2$, we obtain (9).

We can now show (10) on the basis of (9). Suppose that $\gamma_t$ and $\eta_t$ are defined as

$$\gamma_t = 4\log(10dt), \quad \eta_t = \frac{1}{\sqrt{800d\gamma_t^2 + 16\sum_{j=1}^{t-1}\gamma_j^2(\langle \ell_j - m_j, a_j\rangle)^2}}. \tag{32}$$

Denote $\beta_t = 16\gamma_t^2(\langle \ell_t - m_t, a_t\rangle)^2$ and $\eta_t' = \frac{1}{\sqrt{800d\gamma_{t-1}^2 + 16\sum_{j=1}^{t-1}\gamma_j^2(\langle \ell_j - m_j, a_j\rangle)^2}}$. We then have

$$\frac{1}{\eta_{t+1}} - \frac{1}{\eta_t} \geq \frac{1}{\eta_{t+1}'} - \frac{1}{\eta_t} = \sqrt{800d\gamma_t^2 + \sum_{j=1}^t \beta_j} - \sqrt{800d\gamma_t^2 + \sum_{j=1}^{t-1}\beta_j}$$

$$\geq \frac{\beta_t}{2}\left(800d\gamma_t^2 + \sum_{j=1}^t \beta_j\right)^{-1/2} \geq \frac{\beta_t}{4}\left(800d\gamma_t^2 + \sum_{j=1}^{t-1}\beta_j\right)^{-1/2} = \frac{1}{4}\beta_t\eta_t, \tag{33}$$

where the first inequality follows from $\gamma_{t+1} \geq \gamma_t$, the second inequality follows from $\sqrt{y} - \sqrt{y-x} \geq \frac{x}{2\sqrt{y}}$ for $0 \leq x < y$, and the last inequality follows from $\beta_t \leq 800d\gamma_t^2$. We now have

$$\mathbf{E}[R_T(a^*)] \leq d \cdot \mathbf{E}\left[\frac{1}{4}\sum_{t=1}^T \eta_t\beta_t + \frac{\log T}{\eta_T}\right] + 3$$

$$\leq d \cdot \mathbf{E}\left[\sum_{t=1}^{T-1}\left(\frac{1}{\eta_{t+1}} - \frac{1}{\eta_t}\right) + \frac{1}{\eta_{T+1}'} - \frac{1}{\eta_T} + \frac{\log T}{\eta_T}\right] + 3$$

$$= d \cdot \mathbf{E}\left[\frac{1}{\eta_{T+1}} - \frac{1}{\eta_1} + \frac{\log T}{\eta_T}\right] + 3$$

$$\leq 2d\log T \cdot \mathbf{E}\left[\frac{1}{\eta_{T+1}'}\right] = 2d\log T \cdot \mathbf{E}\left[\sqrt{800d\gamma_T^2 + \sum_{t=1}^T \beta_t}\right]$$

$$\leq 32d\log T \cdot \log(10dT) \cdot \mathbf{E}\left[\sqrt{\sum_{t=1}^T(\langle \ell_t - m_t, a_t\rangle)^2 + 50d}\right],$$

where the first inequality follows from (9) and the definition of $\beta_t$, the second inequality follows from (33), and the last inequality follows from (32) and the definition of $\beta_t$. $\square$

## C Construction of a Matrix Satisfying (13)

Here for $C > 1$, a subset $X = \{x_1, \ldots, x_d\} \subseteq \mathcal{A}$ is said to be a *C-barycentric spanner* for $\mathcal{A}$ if every $a \in \mathcal{A}$ can be expressed as a linear combination of elements in $X$ with coefficients in $[-C, C]$.

**Theorem 5** (Proposition 2.4 in [12])**.** *Suppose $\mathcal{A} \subseteq \mathbb{R}^d$ is a full-dimensional compact set. Given $C > 1$ and an algorithm for linear optimization over $\mathcal{A}$, we can compute a C-barycentric spanner for $\mathcal{A}$ in polynomial time, making $O(d^2\log_C d)$ calls to the linear optimization oracle.*

Given a 2-barycentric spanner $X = \{x_1, \ldots, x_d\}$, define $S$ by

$$M = (x_1\ x_2\ \cdots\ x_d), \quad S = MM^\top = \sum_{i=1}^{d} x_i x_i^\top. \tag{34}$$

Recalling that $x_i \in \mathcal{A}$ and $m \in \mathcal{L}$, we have $\|m\|_S^2 = m^\top \left(\sum_{i=1}^{d} x_i x_i^\top\right) m = \sum_{i=1}^{d} (\langle x_i, m\rangle)^2 \leq d$.
Further, from the definition of a 2-barycentric spanner, for any $a \in \mathcal{A}$, there exists $u \in [-2, 2]^d$ such that $a = Mu$ and hence we have $\|a\|_{S^{-1}}^2 = a^\top (M^{-1})^\top M^{-1} a = u^\top u \leq 4d$. Consequently, the matrix $S$ defined by (34) satisfies (13).

*Remark* 2. Given a *d-optimal design* for $\mathcal{A}$, one can construct a matrix $S$ such that

$$\|m\|_S^2 \leq d \text{ for all } m \in \mathcal{L}, \quad \|a\|_{S^{-1}}^2 \leq 1 \text{ for all } a \in \mathcal{A}, \tag{35}$$

which helps improve the regret bound by reducing the term $\log(1 + 4T)$ in Theorem 2 into $\log(1 + T/d)$. Computing *d-optimal design* is, however, harder than computing barycentric spanners for many examples of $\mathcal{A}$.

## D  Proof of Theorem 4

To show Theorem 4, we use the following lower bound:

**Theorem 6.** *Theorem 4.3. in [25], Lemmas 3. and 4. in [35] Suppose that $\ell_t$ is generated as follows: Let $\varepsilon = \min\{\frac{1}{6}, \frac{d}{\sqrt{8T}}\}$ be a fixed parameter. Pick $a^* \in \{-1, 1\}^d$ from a uniform distribution over $\{-1, 1\}^d$. For $t = 1, \ldots, T$, $i(t)$ is chosen from a uniform distribution over $[d]$ independently. Then $\ell_{ti(t)}$ follows a Bernoulli distribution, where $\Pr[\ell_{ti(t)} = 1] = \frac{1 - \varepsilon a_{i(t)}^*}{2}$ and $\Pr[\ell_{i(t)} = -1] = \frac{1 + \varepsilon a_{i(t)}^*}{2}$. For $i \neq i(t)$, $\ell_{ti} = 0$. Then any algorithm for $\mathcal{A} = \{-1, 1\}^d$ suffers regret of*

$$\mathbf{E}\left[\max_{a^* \in \mathcal{A}} R_T(a^*)\right] \geq \frac{\varepsilon T}{2} = \min\left\{\frac{d\sqrt{T}}{\sqrt{32}}, \frac{T}{12}\right\}, \tag{36}$$

*where the expectation is taken over the choice of $\ell_t$ and the randomness of the algorithm.*

Let us prove Theorem 4. Note that $\mathcal{A}$ is given by $\mathcal{A} = \{-1, 1\}^{d-1} \times \{1\}$. Let $\ell_t$ be a random vector defined as follows: for $t \leq L$, $(\ell_{t1}, \ldots, \ell_{t,d-1})$ are generated in a similar way as in Theorem 6 with $T$ and $d$ replaced by $\lfloor L \rfloor$ and $d - 1$, respectively, multiplied by $1/2$, and we set $\ell_{td} = 1/2$. For $t > L$ we set $\ell_t = 0$. We then have $\|\ell_t\|_1 \leq 1$ and $0 \leq \langle \ell_t, a \rangle \leq 1$ for any $a \in \mathcal{A}$ and for $t \leq L$. Combining this and $\ell_t = 0$ for $t > L$, we can confirm that (i), (ii), and (iii) in Theorem 4 hold. From Theorem 6, we have

$$\mathbf{E}\left[\max_{a^* \in \mathcal{A}} R_T(a^*)\right] = \Omega\left(\min\{(d-1)\sqrt{\lfloor L \rfloor}, \lfloor L \rfloor\}\right) = \Omega(d\sqrt{L}), \tag{37}$$

where the last equality follows from the assumptions of $L = \Omega(d^2)$ and $d \geq 1$, and the expectation is taken over the choice of $(\ell_t)_{t=1}^T$ and the randomness of the algorithm. Since (37) holds for the expectation, there is a (fixed) realization of $(\ell_t)_t^T$ for which (37) holds. If we fix $(\ell_t)_t^T$ to them, $\operatorname{argmax}_{a^* \in \mathcal{A}} R_T(a^*)$ has no randomness, and hence we can exchange $\max_{a^* \in \mathcal{A}}$ and $\mathbf{E}$, i.e., we have $\mathbf{E}[\max_{a^* \in \mathcal{A}} R_T(a^*)] = \max_{a^* \in \mathcal{A}} \mathbf{E}[R_T(a^*)]$, which implies that (iv) in Theorem 4 holds.

## E  Proof of Lemma 3

For $t = 0, 1, \ldots, T$, define convex functions $f_t : \mathcal{L} \to \mathbb{R}$ and $F_t : \mathcal{L} \to \mathbb{R}$ as follows:

$$f_0(m) = \frac{1}{2}\|m\|_S^2,$$

$$f_t(m) = \frac{1}{2}(\langle \ell_t - m, a_t\rangle)^2 \qquad (t \in [T]),$$

$$F_t(m) = \sum_{j=0}^{t} f_j(m) \qquad (t \in \{0, 1, \ldots, T\}).$$

Then, the definition (11) of $m_t$ can be rewritten as

$$m_t \in \operatorname*{argmin}_{m \in \mathcal{L}} F_{t-1}(m).$$ (38)

Using this fact recursively, we have

$$F_T(m^*) \geq F_T(m_{T+1}) = F_{T-1}(m_{T+1}) + f_T(m_{T+1}) \geq F_{T-1}(m_T) + f_T(m_{T+1})$$

$$= f_{T-2}(m_T) + f_{T-1}(m_T) + f_T(m_{T+1}) \geq \cdots \geq f_0(m_1) + \sum_{t=1}^{T} f_t(m_{t+1})$$

$$\geq \sum_{t=1}^{T} f_t(m_{t+1})$$

for arbitrary $m^* \in \mathcal{L}$. From this, we have

$$\sum_{t=1}^{T} (\langle \ell_t - m_t, a_t \rangle)^2 - \sum_{t=1}^{T} (\langle \ell_t - m^*, a_t \rangle)^2 = 2 \sum_{t=1}^{T} f_t(m_t) - 2 \sum_{t=1}^{T} f_t(m^*)$$

$$= 2 \sum_{t=1}^{T} f_t(m_t) - 2(F_T(m^*) - f_0(m^*)) \leq 2f_0(m^*) + 2 \sum_{t=1}^{T} (f_t(m_t) - f_t(m_{t+1}))$$

$$= \|m^*\|_S^2 + 2 \sum_{t=1}^{T} (f_t(m_t) - f_t(m_{t+1})).$$ (39)

We next show

$$f_t(m_t) - f_t(m_{t+1}) \leq 4\|a_t\|_{A_t^{-1}}^2$$ (40)

where we define positive semi-definite matrices $A_t \in \mathbb{R}^{d \times d}$ by

$$A_t = S + \sum_{j=1}^{t} a_j a_j^\top$$ (41)

for $t = 0, 1, \ldots, T$. To show (40), we use the fact that $F_t$ is $A_t$-strongly convex, i.e., it holds for any $m, m' \in \mathcal{L}$ that

$$F_t(m') \geq F_t(m) + \langle \nabla F_t(m), m' - m \rangle + \|m' - m\|_{A_t}^2.$$ (42)

Further, (38) implies that

$$\langle \nabla F_{t-1}(m_t), m - m_t \rangle \geq 0$$ (43)

for any $m \in \mathcal{L}$ and $t \in [T]$. From (42) and (43), we can show (40) as follows:

$$f_t(m_t) - f_t(m_{t+1}) = F_t(m_t) - F_t(m_{t+1}) - F_{t-1}(m_t) + F_{t-1}(m_{t+1})$$

$$\leq \langle \nabla F_t(m_t), m_t - m_{t+1} \rangle - \|m_t - m_{t+1}\|_{A_t}^2 + \langle \nabla F_{t-1}(m_{t+1}), m_{t+1} - m_t \rangle$$

$$\leq \langle \nabla F_t(m_t) - \nabla F_{t-1}(m_t), m_t - m_{t+1} \rangle + \langle \nabla F_{t-1}(m_{t+1}) - \nabla F_t(m_{t+1}), m_{t+1} - m_t \rangle$$

$$\quad - \|m_t - m_{t+1}\|_{A_t}^2$$

$$= \langle \nabla f_t(m_t), m_t - m_{t+1} \rangle - \|m_t - m_{t+1}\|_{A_t}^2 - \langle \nabla f_t(m_{t+1}), m_{t+1} - m_t \rangle$$

$$= \langle \nabla f_t(m_t) + \nabla f_t(m_{t+1}), m_t - m_{t+1} \rangle - \|m_t - m_{t+1}\|_{A_t}^2$$

$$\leq \|\nabla f_t(m_t) + \nabla f_t(m_{t+1})\|_{A_t^{-1}} \|m_t - m_{t+1}\|_{A_t} - \|m_t - m_{t+1}\|_{A_t}^2$$

$$\leq \frac{1}{4} \|\nabla f_t(m_t) + \nabla f_t(m_{t+1})\|_{A_t^{-1}}^2 = \frac{1}{4} \|(\langle m_t - \ell_t, a_t \rangle + \langle m_{t+1} - \ell_t, a_t \rangle)a_t\|_{A_t^{-1}}^2 \leq 4\|a_t\|_{A_t^{-1}}^2,$$

where the first and second inequalities follow from (42) and (43) respectively, the third inequality follows from Cauchy–Schwarz inequatlity, the forth inequality follows from the fact that $a^2 - ab + b^2/4 = (a - b/2)^2 \geq 0$ for $a, b \in \mathbb{R}$. Combining (39) and (40), we obtain

$$\sum_{t=1}^{T} (\langle \ell_t - m_t, a_t \rangle)^2 - \sum_{t=1}^{T} (\langle \ell_t - m^*, a_t \rangle)^2 \leq \|m^*\|_S^2 + 8 \sum_{t=1}^{T} \|a_t\|_{A_t^{-1}}^2.$$ (44)

We next show

$$\sum_{t=1}^{T} \|a_t\|_{A_t^{-1}}^2 \leq d \log \left( 1 + \frac{T}{d} \max_{a \in \mathcal{A}} \|a\|_{S^{-1}}^2 \right). \tag{45}$$

Each $\|a_t\|_{A_t^{-1}}^2$ can be bounded by $\log \det A_t - \log \det A_{t-1}$. In fact, we have

$$\begin{aligned}
\log \det A_t - \log \det A_{t-1} &= -(\log \det(A_t - a_t a_t^\top) - \log \det A_t) \\
&= -\log \det(A_t^{-\frac{1}{2}}(A_t - a_t a_t^\top)A_t^{-\frac{1}{2}}) = -\log \det(I - A_t^{-\frac{1}{2}} a_t a_t^\top A_t^{-\frac{1}{2}}) \\
&= -\log(1 - \|A_t^{-\frac{1}{2}} a_t\|_2^2) \geq \|A_t^{-\frac{1}{2}} a_t\|_2^2 = \|a_t\|_{A_t^{-1}}^2,
\end{aligned}$$

where the forth equality holds since the matrix $(I - A_t^{-\frac{1}{2}} a_t a_t^\top A_t^{-\frac{1}{2}})$ has eigenvalues $\lambda_1' = 1 - \|A_t^{-\frac{1}{2}} a_t\|_2^2$ and $\lambda_2' = \lambda_3' = \cdots = \lambda_d' = 1$, and the inequality follows from $\log(1 + y) \leq y$ for $y > -1$. From this, we have

$$\sum_{t=1}^{T} \|a_t\|_{A_t^{-1}}^2 \leq \log \det A_T - \log \det A_0 = \log \det \left( I + \sum_{t=1}^{T} S^{-\frac{1}{2}} a_t a_t^\top S^{-\frac{1}{2}} \right) = \sum_{i=1}^{d} \log(1 + \lambda_i),$$
$$\tag{46}$$

where $\lambda_1, \lambda_2, \ldots, \lambda_d \geq 0$ are eigenvalues of $\sum_{t=1}^{T} S^{-\frac{1}{2}} a_t a_t^\top S^{-\frac{1}{2}}$. Since we have

$$\sum_{i=1}^{d} \lambda_i = \mathrm{tr} \left( \sum_{t=1}^{T} S^{-\frac{1}{2}} a_t a_t^\top S^{-\frac{1}{2}} \right) = \sum_{t=1}^{T} \|a_t\|_{S^{-1}}^2 \leq T \max_{a \in \mathcal{A}} \|a\|_{S^{-1}}^2,$$

the right-hand side of (46) can be bounded as

$$\sum_{i=1}^{d} \log(1 + \lambda_i) \leq d \log \left( 1 + \frac{T}{d} \max_{a \in \mathcal{A}} \|a\|_{S^{-1}}^2 \right) \tag{47}$$

which implies that (45) holds. Combining (44) and (45), we obtain the inequality in Lemma 3.