[Reviews · NeurIPS 2020]

Review 1

Summary and Contributions: This paper provides nearly tight first- and second- order regret bounds for adversarial bandits. They result is achieved by polynomial-time algorithms which only reliy on linear optimization oracles. The main novelty is the distribution truncation technique which is applied to the standard continuous exponential weight algorithm.

Strengths: The result is new, and will be interesting to bandit community. The solution to this problem is also elegant and easy to understand.

Weaknesses: The proposed polynomial-time algorithm is still impratical because of the use of continuous exponential weight. Is it possible to achieve similar results without continuous exponential weight when the number of actions is finite?

Correctness: I believe the result is correct. Their are some clarity issues that makes me not totally sure (please see the Clarity section).

Clarity: There are some details without explanation in the paper: 1. Is the covariance matrix of the truncated distribution S(\tilde{p}_t) always invertible? 2. How to calculate/approximate the inverse of S(\tilde{p}_t) efficiently? If it is approximated, how to argue that it has small error? Both are important for the validity of the algorithm. Please try to address them in the response. Minor: - In Eq. (20), to avoid confusion, please say that this applies Lemma 1 with S(p_t)^{-1/2}x.

Relation to Prior Work: Table 1 provide a clear comparison. However, the bound shown for previous work does not reflect the real bounds that are known: 1. In the last row of the Table 1 when comparing to [40] (Rakhline&Sridharan), I believe their bound should be <\ell_t - m_t, a_t>^2 in the square root but not \|\ell_t - m_t\|^2. This seems to be an intermediate bound in their anslysis. Please see the top of [Page 27, arxiv/1208.3728], and also [Eq.(3), arxiv/1901.10604]. 2. In the second-last row of Table 1, you show the second-order bound of [29] which gets \|.\|_2^2. I'm not sure whether [29] actually gets a better bound, but again, using the techniques developed in [Lemma 6, arxiv/1208.3728], \|.\|_*^2 is already achievable. Please also check the techniques developed in [Section 3, arxiv/1901.10604], which also already gets \|.\|_*^2 dependence for the second-order bound.

Reproducibility: Yes

Additional Feedback:


Review 2

Summary and Contributions: This paper concerns the adversarial linear bandit setting. In this paper new algorithms are derived that have near optimal first and second-order regret bounds. Post-rebuttal: The authors adequately addressed my comments, which I why I raised my score.

Strengths: The main technical novelty in this paper is that instead of sampling from the multiplicative weight distribution the new algorithm samples from a truncated version of the multiplicative weight distribution. Previously the multiplicative weight distribution would need to be mixed with a stationary distribution to ensure that the covariance matrix of the samples would be suitably bounded. By sampling from the truncated distribution the algorithm ensures that the covariance matrix behaves nicely, which in turn ensures that the loss estimates behave nicely. By using the fact that the multiplicative weight distribution is log-concave it is shown that sampling from the truncated distribution is not too harmful for the remainder of the analysis. Combining the nicely behaved loss estimates with the optimistic mirror descent framework leads to the novel first and second-order regret bounds. The last new result in this paper is a lower bound for this adversarial linear bandits which indeed shows that the first and second-order regret bounds almost match the lower bound.

Weaknesses: The improvement upon previous second-order bounds such as of Rakhlin and Sridharan (2013), who use optimistic mirror descent, is from d sqrt(theta x variance) to d sqrt(variance), where theta is the self-concordance parameter. For some sets the self-concordance parameter can be d but for other sets it can be 1, such as for the unit ball. For the unit ball the algorithm of Rakhlin and Sridharan (2013) has significantly smaller runtime than continuous multiplicative weights and the same regret bound. This should also be made clear in the main text, as right now only the sentence “note that the self-concordance parameter can be d” hints at the above. Also note that assumption (iii) is not an assumption due to the existence of the universal barrier.

Correctness: The proofs appear to be correct.

Clarity: The writing is clear. The proof of Theorem 1 is well explained. As a minor negative remark, although Lemma 1 is an important part of the analysis from the main text it does not become clear how Lemma 1 is used, which is a shame as I think that together with Lemma 4 it is one of the main new technical tools used to derive the novel results.

Relation to Prior Work: A similar improvement from optimistic mirror descent to continuous multiplicative weights is observed in Bubeck and Eldan (2015) and Van der Hoeven et. al. (2018), but with their methods second order bounds are out of the question due mixing with a stationary distribution. Also, a similar technique to find the optimal hint was used by Cutkosky (2019) (section 6), although the loss used is slightly different the techniques are alike.

Reproducibility: Yes

Additional Feedback: Minor comments and refferences: Line 128: with Q^* was provided → perhaps change to: where Q^* is known beforehand Equation (31): I think there is a t missing from gamma For consistent O notation (14) should be an equality rather than an inequality Abernethy, J., Hazan, E. E., & Rakhlin, A. (2008). Competing in the dark: An efficient algorithm for bandit linear optimization. In 21st Annual Conference on Learning Theory, COLT 2008 (pp. 263-273). Bubeck, S., & Eldan, R. (2015). The entropic barrier: a simple and optimal universal self-concordant barrier. In Conference on Learning Theory (pp. 279-279). Cutkosky, A. (2019). Combining Online Learning Guarantees. In Conference on Learning Theory (pp. 895-913). Rakhlin, A., & Sridharan, K. (2013). Online Learning with Predictable Sequences. In Conference on Learning Theory (pp. 993-1019). Van der Hoeven, D., Erven, T., & Kotłowski, W. (2018). The Many Faces of Exponential Weights in Online Learning. In Conference On Learning Theory (pp. 2067-2092).


Review 3

Summary and Contributions: Derive first order and second order regret bound for adversarial linear bandits using truncated distribution.

Strengths: If the paper is correct and efficient, I think it would be a breakthrough in the linear bandits literature. It not only provides various common adaptive regret bound, but also matches the optimal dependency on the dimension d. Also another surprising point is that the algorithm is very simple, which uses a common trick to obtain first order regret bounds.

Weaknesses: If the claim is correct and the algorithm is efficient, it looks no evident weakness to me. Also as the authors claimed, the paper is parameter-free, that is, obtaining the claimed bounds without knowing these quantities.

Correctness: At first glance it seems too good to be true as the algorithm is even arguably simpler than [18] and [31], which only get \sqrt{T} regret bound. When checking the correctness of the proof, I found many parts where details are not provided, which makes me hard to verify it. For example, it is known that in MAB, if one uses truncated distribution, then the standard loss estimator is not unbiased anymore. Here I don't see it is trivial to see the estimator is unbiased. Specifically, why is the matrix in (8) always invertible? Another part is (20). It is very mysterious Lemma 1 is used here. The norms are not the same and the condition S(p) <= I is not verified either. Moreover, how to compute S(p')^-1 efficiently to construct the loss estimator is also an issue.

Clarity: Yes, for the language it is great. But for the analysis there are some hand-waving parts. For example, in Line 395, I think the authors should argue more clearly why x the authors refer to is larger than -1. Also the parts in Lemma 4 using Lemma 1 is not well explained.

Relation to Prior Work: It think the method the authors use is more like [6] than [39] and [14], but it is not mentioned in the paper. For example, in Line 395, I think the authors should argue more clearly why x the authors refer to is larger than -1.

Reproducibility: Yes

Additional Feedback: (after the author response phase) I think the authors indeed answered my questions regarding correctness. I am just still not sure about the efficiency and error control. In the paper, the analysis they did are all based on the assumption that they can access accurate S(p_t)^-1, S(~p_t)^-1, etc, which is not the case if we want a polynomial algorithm. I think the algorithm is efficient even when considering error like what did in "Oracle-Efficient Algorithms for Online Linear Optimization with Bandit Feedback", but I still want to know how small epsilon should be chosen to make all these things work. Hope the authors can explain these more clearly in the final version.

[Author Response · NeurIPS 2020]

**Dear Reviewer #1:**

> Is it possible to achieve similar results without continuous exponential weight when the number of actions is finite?

Currently, we have no idea for bypassing continuous exponential weight. As mentioned around Lines 84–88 of the manuscript, any existing algorithm for finite action sets that does not rely on continuous exponential weight mixes $p_t$ with another distribution, which hinders improved first- or second-order regret bounds. As you commented, however, bypassing continuous exponential weight would improve practical computational efficiency, which we consider as an important future work.

> 1. Is the covariance matrix of the truncated distribution $S(\tilde{p}_t)$ always invertible?

Yes, $S(\tilde{p}_t)$ is invertible. This follows from the assumption that $\mathcal{A}$ is not contained in any proper linear subspace, which is stated at Line 258 of the manuscript. Indeed, under this assumption, $\mathcal{A}'$ is a full-dimensional convex set with a positive Lebesgue measure. Combining this and Lemma 1, we can see that the domain of $\tilde{p}_t$ is full-dimensional as well. Therefore, the distribution $\tilde{p}_t$ has a density function taking positive values over a full-dimensional convex set, which implies that $S(\tilde{p}_t)$ is positive-definite. A similar argument can be found, e.g., in p.8 of [Ito et al., oracle-efficient algorithms for online linear optimization with bandit feedback, NeurIPS2019] (between Eq. (4) and (5)), and is implicitly used in [Bubeck, Lee, Eldan (2017)] as well. In the revised manuscript, we add a more clarified proof.

> 2. How to calculate/approximate the inverse of $S(\tilde{p}_t)$ efficiently?

Since $\tilde{p}_t$ is log-concave, for any $\epsilon > 0$, we can get an $\epsilon$-approximation of $S(\tilde{p}_t)$ w.h.p. by generating $(d/\epsilon)^{O(1)}$ samples from $\tilde{p}_t$, from Corollary 2.7 of [Lovàsz and Vempara (2007)]. Samples from $\tilde{p}_t$ can be generated with their polynomial-time sampling algorithm as mentioned in Section 4.4 of our manuscript. A similar discussion can be found in Lemma 5.17 of [Bubeck, Lee, Eldan (2017)] and around Corollary 1 of [Ito et al., oracle-efficient ..., NeurIPS2019]. This fact is implicitly used in [Hazan and Karnin (2016)] as well. We clarify this in the revised manuscript.

> - In Eq. (20), to avoid confusion, please say that this applies Lemma 1 with $S(p_t)^{-1/2}x$.

Yes. We shall state this more clearly in the revised manuscript. For more details, please see the response to Reviewer#2.

**Dear Reviewer #2:**

> For the unit ball the algorithm of Rakhlin and Sridharan (2013) has significantly smaller runtime..

We agree with this comment. In the revised version, we shall note these facts the reviewer pointed out.

> Also note that assumption (iii) is not an assumption due to the existence of the universal barrier.

We guess that the reviewer read the sentence "(iii) $\mathcal{A}$ has a self-concordant barrier with parameter $\theta \geq 1$" as "there exists $\theta \geq 1$ such that $\mathcal{A}$ has a $\theta$-self-concordant barrier." We meant, however, that "for a given $\theta \geq 1$, $\mathcal{A}$ has a $\theta$-self-concordant barrier," which is an assumption on $\theta$ and $\mathcal{A}$.

> from the main text it does not become clear how Lemma 1 is used,

As Reviewer #1 mentioned, we use Lemma 1 for $x = S(p_t)^{-1/2}y$ with $y \sim S(p_t)$. We can see that assumptions in Lemma 1 hold since we have $\mathbf{E}[xx^\top] = S(p_t)^{-1/2}\mathbf{E}[yy^\top]S(p_t)^{-1/2} = S(p_t)^{-1/2}S(p_t)S(p_t)^{-1/2} = I$ and since log-concavity is preserved under any liner transformation. Using Lemma 1 for $x = S(p_t)^{-1/2}y$, we obtain high-probability bounds for $\|x\|_2^2 = \|S(p_t)^{-1/2}y\|_2^2 = \|y\|_{S(p_t)^{-1}}^2$. We add a clear description of this in the revision.

**Dear Reviewer #3:**

> For example, in Line 395, I think the authors should argue more clearly why x the authors refer to is larger than -1.

We can confirm that $x > -1$ holds since $x$ here corresponds to $x = -1 + \mathbf{E}[\exp(-\eta_t\langle \hat{\ell}_t - m_t, x\rangle)]$, as can be seen from the transformation in lines 393–395. We add a more clarified explanation in the revision.

> Also the parts in Lemma 4 using Lemma 1 is not well explained.. Another part is (20).

Please refer to the response for Reviewer #2.

> it is known that in MAB, if one uses truncated distribution, then the standard loss estimator is not unbiased anymore.

The unbiasedness is proved in the proof of Lemma 2. We guess that the standard loss estimator the reviewer refers to is the one using $S(p_t)^{-1}$ instead of $S(\tilde{p}_t)^{-1}$. This "standard" one may be indeed biased as the reviewer pointed out.

> Specifically, why is the matrix in (8) always invertible? ... Moreover, how to compute $S(p')^{-1}$ efficiently ...

Please refer to the response for Reviewer #1.

[Meta-Review · NeurIPS 2020]

Reviewers are satisfied with the rebuttal, and two of them increased their score. The results are strong, but please do take all reviewers' comments into account and improve the writing for the final version, especially in terms of the invertibility of S(\tilde{p}_t) and the efficiency of the algorithm.